# Measurement of Synergy Degree between Environmental Protection and Industrial Development in the Yellow River Basin and Analysis of Its Temporal and Spatial Characteristics

Weixian Xue and Yunru Liu *

School of Economics and Management, Xi'an University of Technology, Xi'an 710000, China
* Correspondence: yunru-liu@stu.xaut.edu.cn

**Abstract:** The Yellow River Basin plays an important role in economic and social development and ecological security; therefore, its ecological protection and high-quality development are vital. In order to understand the level of synergistic development between environment and industry in the Yellow River Basin, and to understand the change in synergy degree through spatial and temporal analysis, and finally to propose suggestions to provide a basis for the policy formulation of environmental protection and industrial development, informing the initiatives of the relevant parties—companies and residents, so as to ensure the high-quality development and sustainable development of the Yellow River Basin, this paper is based on the theory of synergetic and dissipative structure, and it expounds the synergetic mechanism by constructing the compound system of environment and industry in the Yellow River Basin and revealing their internal and external interactions. Based on the panel data of 57 prefecture-level cities in the Yellow River Basin from 2010 to 2020, the synergy degree of environment and industry in the Yellow River Basin and its temporal and spatial characteristics are discussed by using the synergy model of compound systems. The results show that: (1) the overall degree of environmental and industrial synergism in the Basin develops from mildly non-synergistic to mildly synergistic, but the level is still low; there are significant temporal and regional differences in synergy degree in the upper, middle and lower reaches of the Basin and among cities. (2) The number of cities in the basin that are in mild synergy is increasing; the synergy degree shows an overall positive global spatial autocorrelation.

**Keywords:** Yellow River basin; environmental protection; industrial development; synergy mechanism; synergy degree model

## 1. Introduction

Ecological protection and high-quality development in the Yellow River Basin have become a major national strategy, and the synergistic development of ecological environment and industry is an important driver of this strategy. However, the synergistic development of ecological environmental protection and industry in the Yellow River Basin still faces the real dilemma of unreasonable industrial structure, strong homogeneity of leading industries, weak competitiveness and fragile ecological environment [1]. It is still necessary to measure the degree of synergy to determine the level of synergy and how to promote the synergy.

Domestic and foreign scholars mainly study the synergy of the two from the perspective of synergy mechanism and synergy degree measurement. In the few research studies on the synergistic mechanism of the two, the interaction is mainly studied. For example, Weng Gangmin et al. (2021) [2] discussed the synergy mechanism by analyzing the complementary and mutually restrictive relationship among tourism, ecology and urbanization. Jiang Y H et al. (2021) [3] designed a theoretical framework for the collaborative evolution of logistics industry and ecological environment based on the relationship between the two. Among the studies on the measurement of synergy between the two, from the perspective

of research, one focuses on the theory of system science. For example, Cao Honghua et al. (2018) [4] took Erhai River Basin as the object to build the comprehensive evaluation and coupling evolution model of agriculture—ecosystems. The other focuses on the perspective of synergism, mainly focusing on the degree of synergy between tourism industry and environment. For example, Xiang Li (2017) [5] studied the coupling and coordinated development of tourism industry and ecological environment, and drew the conclusion that tourism should pay particular attention to the coordination with ecological environment. Gelso B R and Peterson J M (2005) [6] used quantitative methods such as the environmental Kuznets (EKC) curve to analyze the coordinated development of tourism and environmental systems. In terms of research methods, the coupled coordination degree model, Haken model [7], compound system synergy model [8], Lotka-Volterra model, system evolution and system dynamics model are mostly used. Among them, the coupled coordination degree model is used more often.

To sum up, on the one hand, most of the existing research on the synergistic mechanism between environment and industry, we have studied the interactive relationship between the two, while few have fully and completely expounded the synergistic mechanism. Based on the theory of synergetic and dissipative structure, this paper analyzes whether the environmental and industrial compound system in the Yellow River Basin meets the synergetic conditions on the basis of constructing the compound system. Through the interaction between the system and the external environment and the nonlinear interaction of the sub-systems dominated by order parameters, the mechanism of the synergy is explained, and the whole process of the synergy is explained theoretically. On the other hand, for the measurement methods of the synergy degree between environment and industry, the coupled coordination degree model is used more often than the compound system synergy model. The latter is more suitable for constructing the compound system and finding out the degree of synergy of order parameters. This paper brings this approach to a wider use in the environment and industry at large.

## 2. Synergistic Mechanism between Environment and Industry in the Yellow River Basin

In this section, the synergistic mechanism between environment and industry in the Yellow River basin is the theoretical basis for the construction of the composite system synergy model, because the composite system synergy model needs to be built based on the sequential parametric principle and the servitude principle of the synergist for the constructed composite system. The clarification of the sequential parametric principle and the servitude principle requires the study of how an open system, under the condition of material, energy and information exchange with the external environment (this condition is represented by the control parameter), can self-organize through the nonlinear interactions between subsystems when the control parameter reaches a critical value, and finally the system evolves to an orderly one under the domination of the sequential parametric principle [9,10]. Therefore, this section differs from the previous mechanism of describing only the interaction between environment and industry. It will firstly construct the composite system; secondly analyze and confirm that it satisfies the composite system synergy condition. Finally, the self-organized evolution process is fully described through the exchange process between the system and the external environment and the nonlinear interaction of subsystems under the domination of the order parameter.

### 2.1. Structural Analysis of Environmental and Industrial Compound System in Yellow River Basin

The environmental and industrial compound system in the Yellow River Basin is composed of environmental and industrial subsystems with different attributes. Under the premise of material, energy and information exchange with external environment such as policy, the two subsystems interact and penetrate each other. The specific structure is shown in Figure 1, where the dotted line of the ellipse is the system boundary; the system itself is inside the system boundary, including two subsystems: the environmental subsystem and

the industrial subsystem; and the environment related to the system is outside the system boundary, including policy environment, economic environment and social environment.

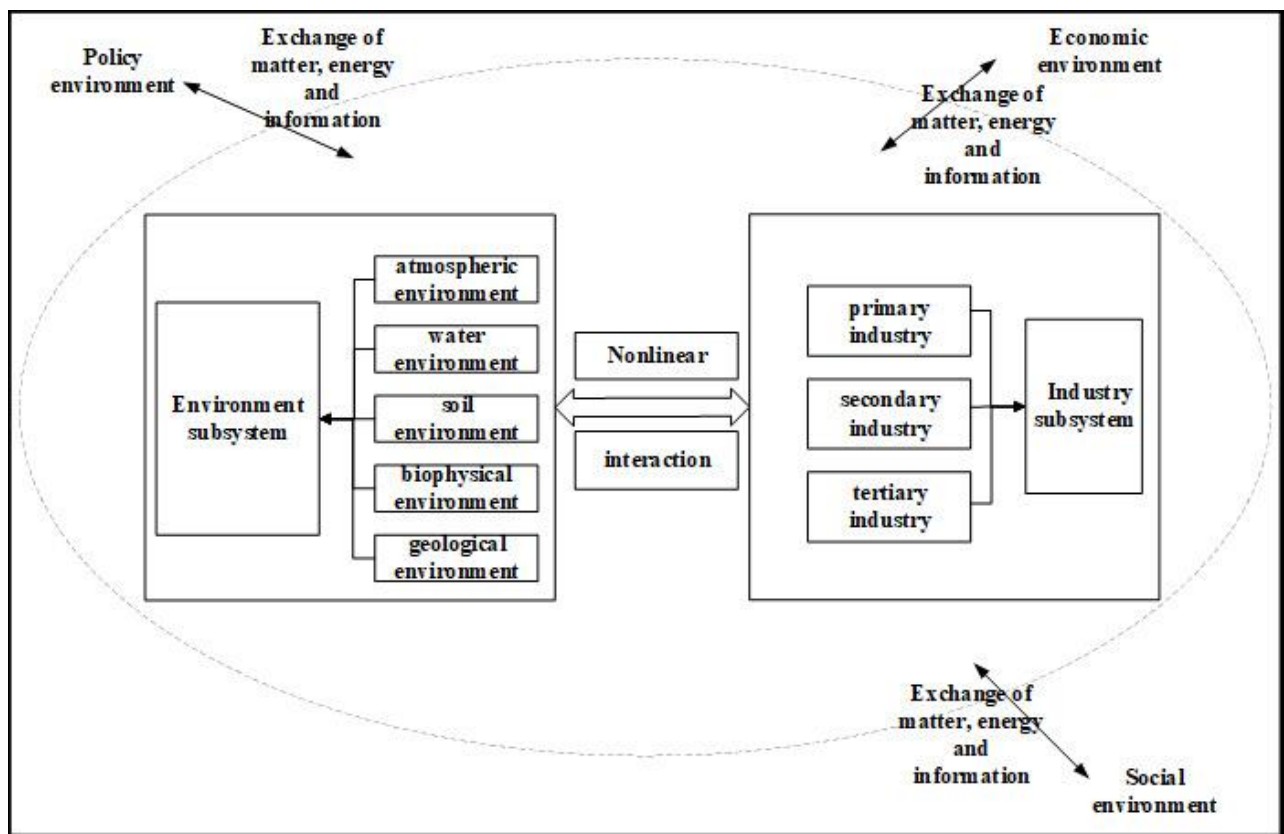

**Figure 1.** Boundary and composition of environment and industry compound system.

Specifically, the study in this paper focuses on the natural environment, so the environmental subsystem of the Yellow River Basin is considered to include the atmospheric environment, water environment, soil environment, biological environment and geological environment of the Yellow River Basin in five parts [11]. The industrial subsystem namely includes the primary, secondary and tertiary industries of the Yellow River Basin. The policy environment is broader, mainly government policies, such as environmental regulation policies, or policies developed in response to unexpected situations such as COVID-19. The economic environment includes various factors such as consumers' income level, consumer spending pattern and consumption structure, consumer savings and credit, economic development level, and urbanization level. The social environment mainly includes innovation drive, livelihood and welfare, and safety and security.

The environmental subsystem and the industrial subsystem are closely linked through capital flows (funds or resources) to achieve nonlinear interactions. The main body of the industry-enterprise needs to input production factors and resources in the process of production and produce pollutants in the process, which requires the ecological environment to provide factor endowment, absorb and degrade production pollutants, i.e., the environmental subsystem provides accommodation space and material basis for the industrial subsystem. At this point, capital flows from the environmental subsystem to the industrial subsystem. The input of the ecological environment to the industrial development is realized in the form of strong investment in environmental protection brought by the industrial development—environmental protection funds, etc., to ensure the sustainable development of environmental input. The environmental subsystem and the industrial subsystem can achieve the goal of spiral upward through positive feedback. The environmental subsystem that exceeds the carrying capacity makes the industrial subsystem lack of resources and

the basic conditions for development unstable, which affects the sustainable development of the industry. This cycle will form a vicious circle and degrade the environmental and industrial complex system to a lower level. Thus, it can be seen that the environmental subsystem and the industrial subsystem are embedded in parallel, and the two together constitute an organic whole that interacts, interpenetrates and constrains each other.

### 2.2. Analysis on the Synergistic Conditions of Environmental and Industrial Compound System in the Yellow River Basin

The environmental and industrial compound system in the Yellow River Basin has the conditions of co-evolution from disorder to order, that is, it is open, far from the equilibrium state, nonlinear interaction and fluctuation of the system [12]. First, openness means that the system can realize the exchange of material, information and energy with the external environment in the process of self-development, which is the premise of the synergy and order of the system. The environmental and industrial compound system in the Yellow River Basin is affected by economic, social and policy environment in the process of development, and its orderly development can provide technical capital and other factors to the external environment. Second, away from the equilibrium state, refers to the system from a long-term without any change in the state of transformation. The Yellow River Basin environmental and industrial compound system has material exchange with the outside world, and there are state transitions. Third, nonlinear action refers to the fact that the interaction between subsystems is not simple linear addition, which is the driving force behind the synergy and orderliness of the system. There are complex positive and negative feedbacks between environmental and industrial subsystems in the Yellow River Basin. Fourth, the fluctuation of the system refers to the deviation of the system from the stable state, which is the incentive of the system synergy. When the system is in the nonlinear region far from the equilibrium state, the random fluctuation will be amplified rapidly through the nonlinear action and chain effect, thus forming the overall huge fluctuation, and causing the system abrupt change [13]. When the compound system of environment and industry in the Yellow River Basin is in the nonlinear region far from the equilibrium state, the small fluctuations of the industrial structure upgrading and rationalization will be amplified by the nonlinear effect of the system, and finally the system will change abruptly and evolve from disorder to order.

### 2.3. Self-Organizing Evolution of Environmental and Industrial Compound System in the Yellow River Basin

First of all, the interaction between the compound system and the external environment is explained. First, the compound system needs to obtain production information, capital and other material support from the economic environment, while the compound system provides capacity for the economic environment. Second, the compound system needs to obtain human resources and infrastructure guarantee from the social environment, while the compound system can provide the social environment with the talents who master advanced technologies and the funds to support the infrastructure. Third, the compound system needs to obtain production and environmental protection information and various development policies from the policy environment, and the compound system affects the formulation of policies such as developing emerging industries.

Secondly, order parameters are determined. According to the definition and characteristics of order parameters, the classification of industrial economics research perspective by Yang Gongpu et al. (2005) [14] and the existing research [15], environmental state, environmental status, environmental governance and industrial structure, industrial agglomeration and industrial competitiveness are determined as the compound system order parameters.

Finally, it is shown that the order parameters govern the nonlinear interactions between the subsystems. Figure 2 shows this. Environment has bearing and a restricting effect on industry. On the one hand, the environmental subsystem provides the material basis, such as factor resources and the accommodation space including absorption and degradation for the industrial subsystem. Environmental subsystem plays a supporting

role to the industry under the control of environmental state, status and governance. Under the control of the stable and orderly environmental state, the environmental carrying capacity is in a high state, which provides lasting factor support and broad space for industrial development. Under the status that the environment can absorb and degrade the emissions brought by industrial development, it can provide carrying capacity for the industry. Under the control of efficient governance, environmental quality will be greatly improved, thus guaranteeing and supporting the green development of industries. On the other hand, the carrying capacity of the environment is limited. Once the negative impact of industrial development on the environment exceeds the threshold of environmental carrying capacity, the development of industrial subsystems will be hindered by the huge damage to the environment. The restriction effect of environmental subsystem on industrial subsystem comes into being under the control of environmental state, status and governance. The environmental state beyond the carrying capacity has deteriorated significantly, which cannot provide material support for industrial development. The environment status is at a poor level, unable to fully absorb the emissions of industrial development; environmental governance has a long-term lag effect, limiting the development of the industry. Industry can promote and stress the environment. On the one hand, the industrial subsystem not only provides funds and technologies for the environmental subsystem, but also its transformation and upgrading reduces resource consumption, which is conducive to environmental protection. Specifically, the industrial subsystem plays a promoting role to the environmental subsystem under the domination of industrial structure, industrial agglomeration and industrial competitiveness. The advanced industrial structure can help reduce pollutant emission, and the rationalization of industrial structure can promote the rational allocation and flow of factors, so as to improve production efficiency and create more wealth, and provide capital and technical support for environmental protection. The rational play of industrial agglomeration effect can promote the diffusion of technology and knowledge in the agglomeration area, improve the output level, and produce positive benefits to the environment. The improvement of industrial competitiveness is reflected in the improvement of industrial innovation and technology, so as to provide technical support for environmental protection. On the other hand, excessive exploitation of the environment in the process of industrial development will damage the environment. The over-exploitation of environment in industrial development is reflected through the domination of industrial structure, industrial agglomeration and industrial competitiveness. The incompatibility of industrial structure intensifies the demand for ecological environment, leading to the deterioration of environmental state and environmental status. When the industrial agglomeration effect is not played effectively, the synergistic economic benefits of industrial agglomeration cannot be reflected, and it cannot provide financial and technical support for the development of environmental subsystems. When the industrial competitiveness is weak, the scientific and technological innovation ability of enterprises is poor, which cannot provide support for the orderly development of environmental subsystems.

From the mechanism of synergistic development of environment and industry, it is clear that environment and industry promote each other and develop together. Therefore, the environmental subsystem and the industrial subsystem cannot be separated, and the joint development of the two can promote the synergistic development of the whole system, forming the overall benefit of "1 + 1 > 2". Therefore, it is convenient to build a model to measure the synergy degree between environment and industry, so that relevant government departments can monitor the synergy between the two and provide a basis for the policy formulation of environmental protection and industrial development, and can inform the initiatives of the relevant parties—companies and residents.

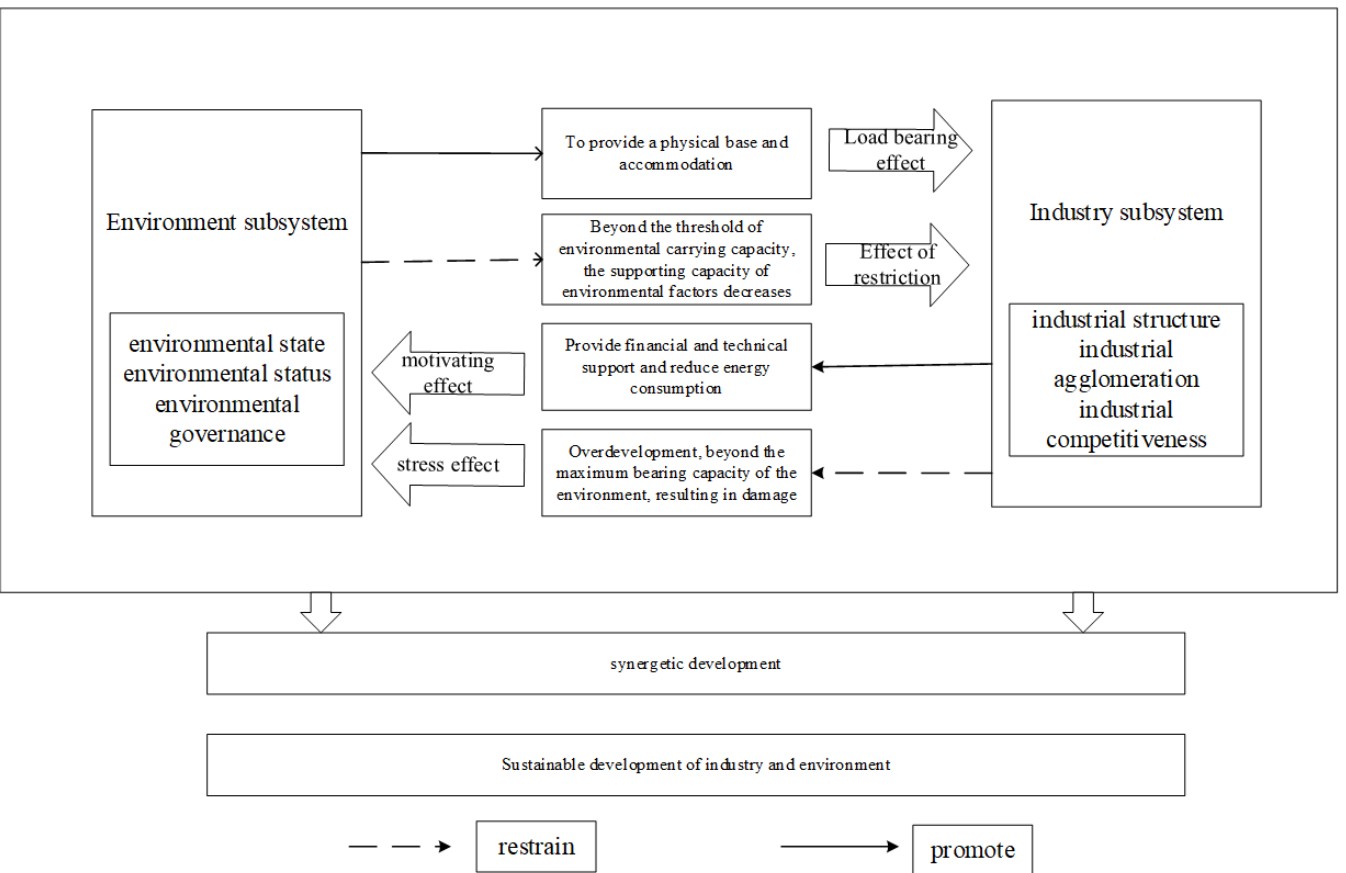

**Figure 2.** Environment and industry synergy map.

## 3. Materials and Methods

### 3.1. Research Methodology

3.1.1. Synergy Model of Compound Systems

This paper draws on the study of Meng Qingsong and Han Wenxiu (2000) [16] to construct a model for measuring the synergy degree between environment and industry. First, the order parameter component orderliness model of subsystem is constructed. The environment and industry are considered as a compound system $S = \{S_1, S_2\}$, where $S_1$ is the environment subsystem and $S_2$ is the industry subsystem. Considering the subsystem $S_j, j \in [1, 2]$, let the order parameter components of the subsystem development process be $e_j = (e_{j1}, e_{j2}, \ldots\ldots, e_{jn})$, where $n \geq 1$, $U_{ji} \leq e_{ji} \leq T_{ji}$, $i = 1, 2, \ldots, n$, $T_{ji}$ and $U_{ji}$ represent the maximum and minimum values of $e_{ji}$, respectively. Here it is assumed that $e_{j1}, e_{j2}, \ldots\ldots, e_{jl1}$ are the positive indicators, i.e., the larger the value, the higher the orderliness of the order parameter components; it is assumed that $e_{jl_1+1}, e_{jl_1+2}, \ldots, e_{jn}$ are the inverse indicators, i.e., the larger the value, the lower the orderliness of the order parameter components. This is shown in the following equation:

$$u_j(e_{ji}) = \begin{cases} \dfrac{e_{ji} - U_{ji}}{T_{ji} - U_{ji}}, i \in [1, l_1] \\ \dfrac{T_{ji} - e_{ji}}{T_{ji} - U_{ji}}, i \in [l_1 + 1, n] \end{cases} \tag{1}$$

where $u_j(e_{ji}) \in [0, 1]$, The larger the value of $u_j(e_{ji})$, the greater the "contribution" of the order parameter component $e_{ji}$ to the ordering of the system.

Next, the orderliness of the order parameter components is integrated in a linear weighted summation to construct a subsystem orderliness model. The details are as follows.

$$u_j(e_j) = \sum_{i=1}^{n} \lambda_i u_j(e_{ji}), \lambda_i \geq 0, \sum_{i=1}^{n} \lambda_i = 1 \tag{2}$$

where $u_j(e_j) \in [0,1]$, The larger the $u_j(e_j)$, the higher the degree of subsystem order and vice versa. The weight coefficient $\lambda_i$ is calculated by the CRITIC assignment method.

Finally, for a given initial moment $t_0$, let the orderliness of each subsystem be $u_j^0(e_j)$, j = 1,2,...,k, at the moment $t_1$ of the system evolution process, the degree of order of each subsystem is $u_j^1(e_j), j = 1, 2, \ldots, k$, define cm as the degree of compound system synergy, which is calculated as follows.

$$cm = \theta \sqrt[k]{\left| \prod_{j=1}^{k} [u_j^1(e_j) - u_j^0(e_j)] \right|}$$
$$\theta = \frac{\min_j[u_j^1(e_j) - u_j^0(e_j) \neq 0]}{\left| \min_j[u_j^1(e_j) - u_j^0(e_j) \neq 0] \right|}, j = 1, 2, \ldots, k \tag{3}$$

where cm $\in [-1,1]$, the higher the value indicates the higher the level of compound system synergy development, and vice versa. The division of the level of compound system synergy degree is shown in Table 1 [17].

**Table 1.** Table of synergy level division of compound system.

| Synergy Degree | Synergy Level |
|---|---|
| cor $\in [-1, -0.666]$ | Highly non-synergistic |
| cor $\in [-0.666, -0.333]$ | Moderately non-synergistic |
| cor $\in [-0.333, 0]$ | Mildly non-synergistic |
| cor $\in [0, 0.333]$ | Mildly synergistic |
| cor $\in [0.333, 0.666]$ | Moderately synergistic |
| cor $\in [0.666, 1]$ | Highly synergistic |

### 3.1.2. Global Spatial Autocorrelation

In this study, the global spatial autocorrelation of synergy degree is judged by constructing the adjacent spatial weight matrix and calculating the global Moran's I index. This index can indicate the spatial agglomeration of environmental and industrial synergy degree in the Yellow River Basin as a whole, and indicate whether there is spatial autocorrelation of synergy degree. Its calculation formula is:

$$I = \frac{\sum_{i=1}^{n} \sum_{j=1}^{n} w_{ij}(x_i - \overline{x})(x_j - \overline{x})}{S^2 \sum_{i=1}^{n} \sum_{j=1}^{n} w_{ij}} \tag{4}$$

where $n$ is the number of regions; $w_{ij}$ is the spatial weight matrix; $x_i$ and $x_j$ is the attribute value of region $i$ and region $j$, respectively, i.e., the synergy degree between environment and industry in each city; $\overline{x}$ is the sample mean; $S^2$ is the sample variance; $I$ is the Moran's I index, with values ranging from $[-1, 1]$, a positive value indicates a positive spatial correlation, a negative value indicates a negative spatial correlation, and a value of 0 indicates no spatial correlation [15]. The $p$-value and $z$-value in the paper are the statistics to test the significant level of Moran's I.

### 3.2. Indicator System and Data Sources

While considering the order parameter of environmental state, environmental status and environmental governance, industrial structure, industrial agglomeration and industrial competitiveness proposed in the synergistic mechanism, the order parameter

components are constructed with reference to the existing research results [15,18–20], as shown in Table 2.

**Table 2.** An index system for measuring the synergy degree of environmental and industrial compound system.

| Subsystems | Order Parameter | Order Parameter Component (Indicators) | Indicator Units and Properties |
|---|---|---|---|
| Environment Subsystem (E) | Environmental State (E1) | Green space per capita (E11) | $m^2$ (+) |
| | | Greening coverage of built-up areas (E12) | % (+) |
| | | Total water supply (E13) | 10,000 $m^3$ (+) |
| | Environmental Status (E2) | Industrial wastewater discharge (E21) | 10,000 tons (−) |
| | | Industrial $SO_2$ emissions (E22) | ton (−) |
| | | Industrial smoke (dust) emissions (E23) | ton (−) |
| | Environmental Governance (E3) | Centralized treatment rate of sewage treatment plants (E31) | % (+) |
| | | Harmless disposal rate of domestic waste (E32) | % (+) |
| Industry Subsystem (S) | Industry Structure (S1) | Industrial structure advanced index (S11) | (+) |
| | | Industrial structure rationalization index (S12) | (−) |
| | Industry Agglomeration (S2) | Zone entropy (S21) | (+) |
| | Industry Competitiveness (S3) | Number of students enrolled in higher education (S31) | the (+) |
| | | Total profit of industrial enterprises above the scale (S32) | 10,000 yuan (+) |
| | | Total import and export (S33) | 10,000 dollars (+) |

Specifically, the selection of each indicator is based on the following: the environmental state should mainly include indicators that characterize the quality of the environment and the state of natural resources, so two indicators are selected, namely, the green space per capita and the greening coverage of built-up areas, while the total water supply is a representative indicator that reflects the characteristics of the Yellow River Basin, so this indicator is added to the indicators of the environmental state; the environmental status should mainly include the pressure brought by human activities. The main pressure on the ecological environment comes from industrial activities after the transition from the agricultural era to the industrial era, and the polluting industries in the Yellow River Basin are concentrated in the secondary industry, which emits a large amount of "three waste" pollutants. Therefore, under the sequential parameter of environmental status, based on the characterization of the indicators of the three pollutants of environmental status in existing studies [21,22], and based on the availability of data, three indicators of industrial wastewater discharge, industrial $SO_2$ emissions and industrial smoke (dust) emissions of environmental status were selected; environmental governance should mainly include measures that are conducive to environmental protection, and based on the availability of data, the centralized treatment rate of sewage treatment plants and the harmless disposal rate of domestic waste are selected to cover production and life. Industrial structure is characterized by the most representative industrial structure advanced index and industrial structure rationalization index; industrial agglomeration is characterized by the most commonly used index of agglomeration—zone entropy; industrial competitiveness is characterized by the number of students enrolled in higher education—the number of education resources related to each industry, the total profit of industrial enterprises above

the scale—the industrial efficiency, and the total import and export—the international competitiveness of industries.

Based on data availability and with reference to the Yellow River Journal Volume 2 [23], 57 cities in eight provinces in the Yellow River Basin (only Aba Tibetan Autonomous Prefecture in Sichuan is involved and data are missing, so it is not considered) are studied in this paper, covering the period of 2010–2020. The data of the indicators are mainly obtained from the China Urban Statistical Yearbook, China Urban Construction Statistical Yearbook, provincial and municipal statistical yearbooks, statistical bulletins on national economic and social development, and information from statistical bureaus. Individual missing data are supplemented by interpolation method.

## 4. Results and Discussion

First, the data were processed by the mean-standard deviation method, and then CRITIC weights were assigned to the standardized data to obtain the weights of each order parameter component, as shown in Table 3.

**Table 3.** The order parameter components weights of environmental subsystem and industrial subsystem.

| Subsystems | Order Parameter Components | Weighting Factor (w) |
| --- | --- | --- |
| Environmental Subsystem | E11 | 0.122 |
| | E12 | 0.112 |
| | E13 | 0.126 |
| | E21 | 0.117 |
| | E22 | 0.132 |
| | E23 | 0.139 |
| | E31 | 0.126 |
| | E32 | 0.126 |
| Industry Subsystem | S11 | 0.176 |
| | S12 | 0.204 |
| | S21 | 0.193 |
| | S31 | 0.139 |
| | S32 | 0.155 |
| | S33 | 0.133 |

By substituting the standardized data into Equation (1), we can obtain the orderliness of order parameter components; combined with the index weights, we can obtain the subsystem orderliness by substituting into Equation (2) and weighting the sum; taking 2010 as the base period, we can obtain the synergy degree by substituting the subsystem orderliness results into Equation (3).

### 4.1. Time Evolution Analysis

The overall environmental and industrial synergy degree in the Yellow River Basin from 2011 to 2020 showed a first decline and then an increase, as shown in Figure 3. The decline of the compound system synergy degree in 2012 mainly lies in the decline of the industrial subsystem orderliness. The main possible reason for the change in industrial subsystem orderliness is that the profits of the enterprises above the scale in more cities in the basin dropped significantly due to the poor macro environment, overcapacity and weak demand.

The overall trend of environmental and industrial synergy degree in the upper, middle and lower reaches of the Yellow River basin is fluctuating upward, but the level of synergy is low and mild. The overall level of upstream remains low. The midstream continues to increase the area of parkland, improve the centralized treatment rate of sewage treatment plants and the harmless treatment rate of domestic waste, reduce industrial waste emissions, and improve the level of production specialization, and the degree of synergy appears to increase significantly. The lower reaches development is at a higher level in the basin overall, but the synergy degree declined slowly after 2017 under the influence of the significant

decrease in profits of the enterprises above the scale and the weaker specialization division of labor. In-depth view of the reasons for the decline, cities in lower reaches of Henan Province may be due to the decline in profits of some key industries in its municipalities, so the profits of the enterprises above the scale decreased; cities of Shandong Province, the reason for the decrease in profits of the enterprises above the scale and weaker specialization division of labor mainly lies in 2018 and 2019 is the starting year of the conversion of old and new dynamic energy in Shandong Province, and facing the fourth economic census, new dynamic energy cultivation and old dynamic energy elimination pressure.

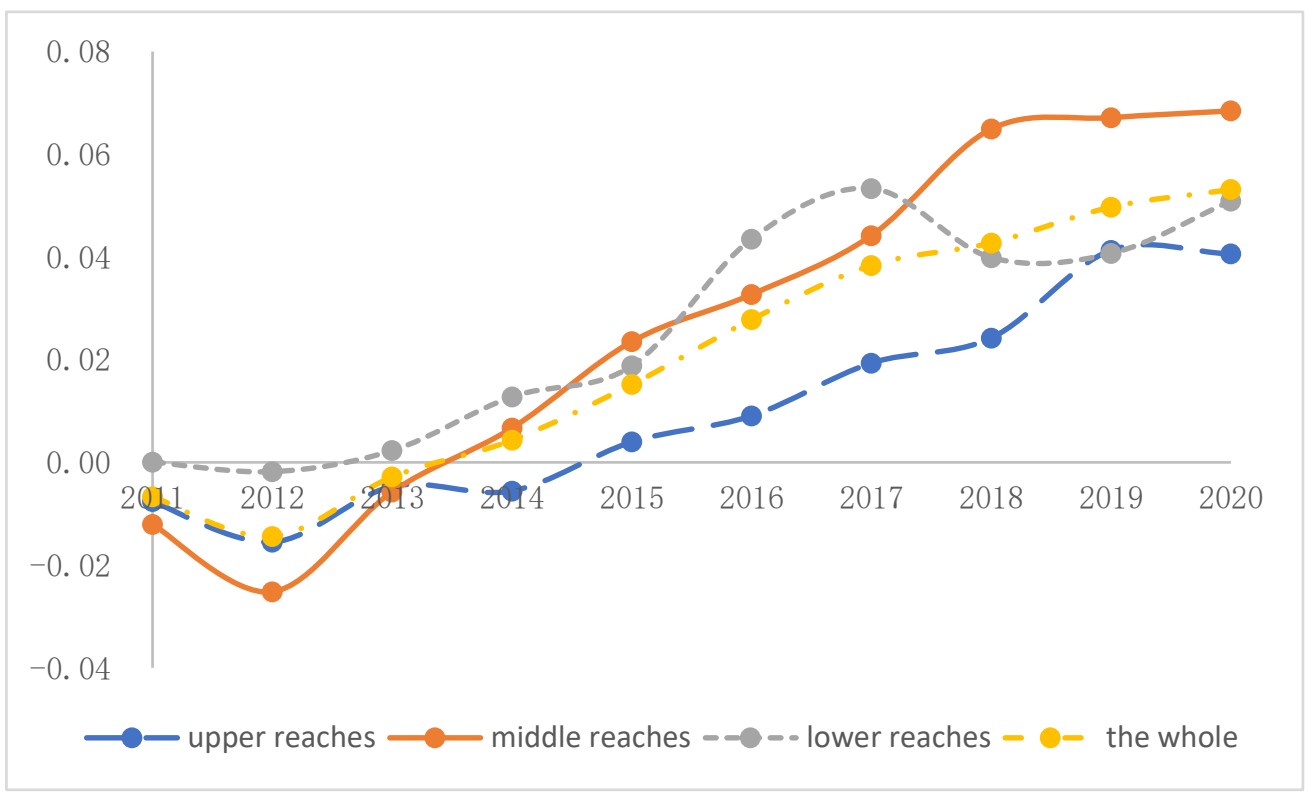

**Figure 3.** Synergy degree of the upper, middle and lower reaches of the Yellow River Basin and the overall compound system from 2011 to 2020.

There are obvious temporal and regional differences in the environmental and industrial synergy degree among cities in the Yellow River basin. Select the representative years of 2011, 2016 and 2020 to draw the radar map of synergy, as shown in Figure 4. The synergy degree of cities widens with time, which shows that the coverage of radar map expands; At the same time, the radar chart changes in the same direction in different years. However, overall, the level of synergy degree in most cities is steadily increasing. Individually, Qingyang, Xi'an and Zhengzhou have significantly higher environmental and industrial synergy degree than other cities in 2016 and 2020. Although Qingyang has a lower level of industrial development, its industrial structure is more reasonable, its industrial subsystem orderliness is higher than the base period in almost all years, its pollutant emissions are minimal, and its sewage treatment plants have a high centralized treatment rate, so the synergy level is relatively high. The reason for the high degree of synergy in the latter two cities is that their industrial and economic development levels are higher, and they influence the environmental subsystem through non-linear interactions, which makes the overall degree of synergy relatively high, but it is still a light level of synergy.

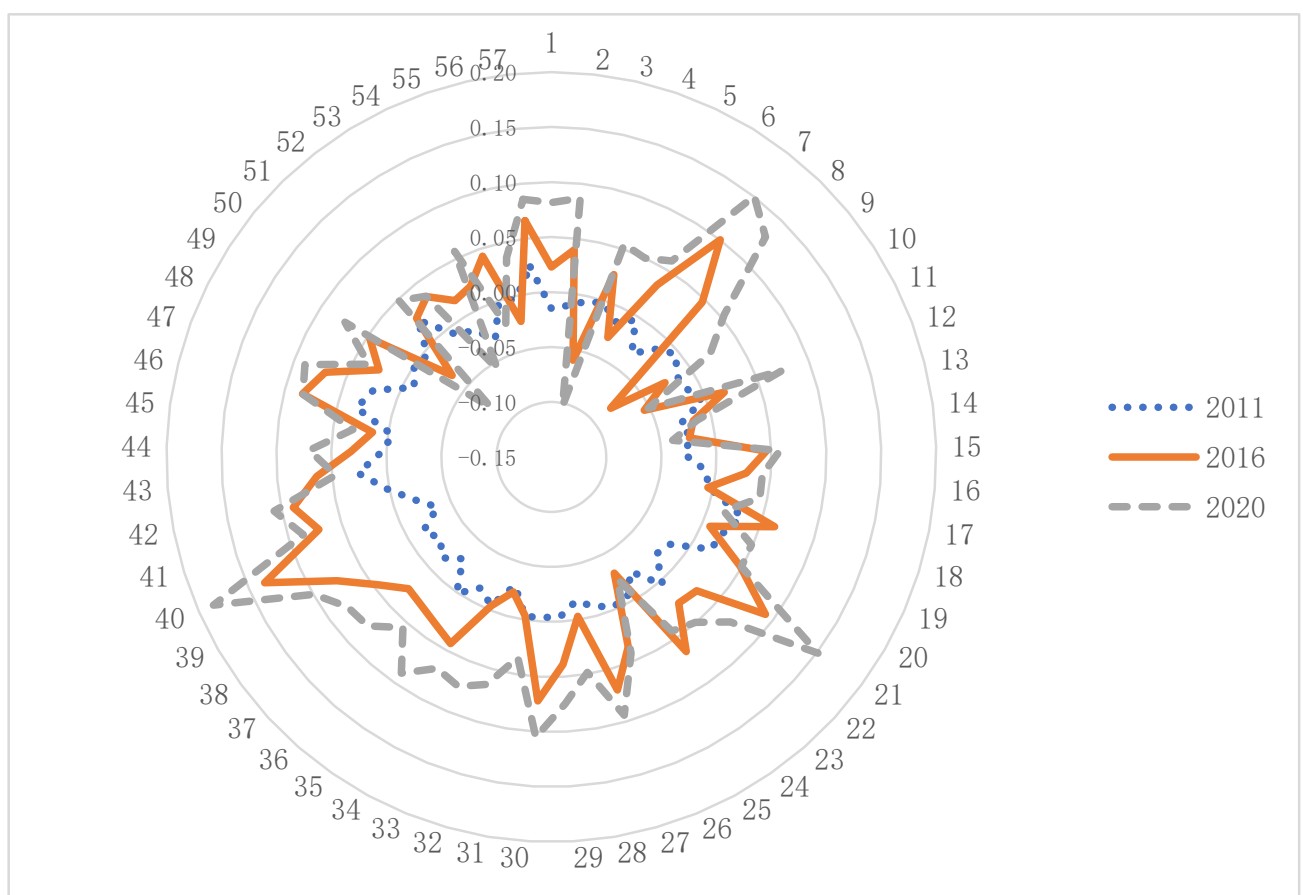

**Figure 4.** Radar map of environmental and industrial synergies in the Yellow River Basin in typical years. Note: The numbers in the figure correspond to the prefecture-level cities in the Yellow River Basin: 1—Xining, 2—Lanzhou, 3—Baiyin, 4—Tianshui, 5—Wuwei, 6—Pingliang, 7—Qingyang, 8—Dingxi, 9—Longnan, 10—Yinchuan, 11—Shizuishan, 12—Wuzhong, 13—Guyuan, 14—Zhongwei, 15—Hohhot, 16—Baotou, 17—Wuhai, 18—Ordos, 19—Bayannur, 20—Ulaanchab, 21—Xi'an, 22—Tongchuan, 23—Baoji, 24—Xianyang, 25—Weinan, 26—Yan'an, 27—Yulin, 28—Shangluo, 29—Taiyuan, 30—Datong, 31—Yangquan, 32—Changzhi, 33—Jincheng, 34—Shuozhou, 35—Jinzhong, 36—Yuncheng, 37—Xinzhou, 38—Linfen, 39—Lvliang, 40—Zhengzhou, 41—Kaifeng, 42—Luoyang, 43—Anyang, 44—Hebi, 45—Xinxiang, 46—Jiaozuo, 47—Puyang, 48—Sanmenxia, 49—Jinan, 50—Zibo, 51—Dongying, 52—Jining, 53—Tai'an, 54—Dezhou, 55—Liaocheng, 56—Binzhou, 57—Heze.

### 4.2. Spatial Evolutionary Analysis

#### 4.2.1. Spatial Divergence Analysis

In order to analyze the spatial variation of environment–industry synergy development of cities in the Yellow River Basin more comprehensively, we selected the synergy degree in 2011, 2016 and 2020 as the evaluation objects, and used ArcGIS 10.7 software to draw the spatial pattern distribution of synergy degree of each city, which is shown in Figures 5–7. The number of cities with light synergy between environment and industry in the Yellow River Basin is increasing. This indicates that the basin as a whole is evolving in an orderly direction.

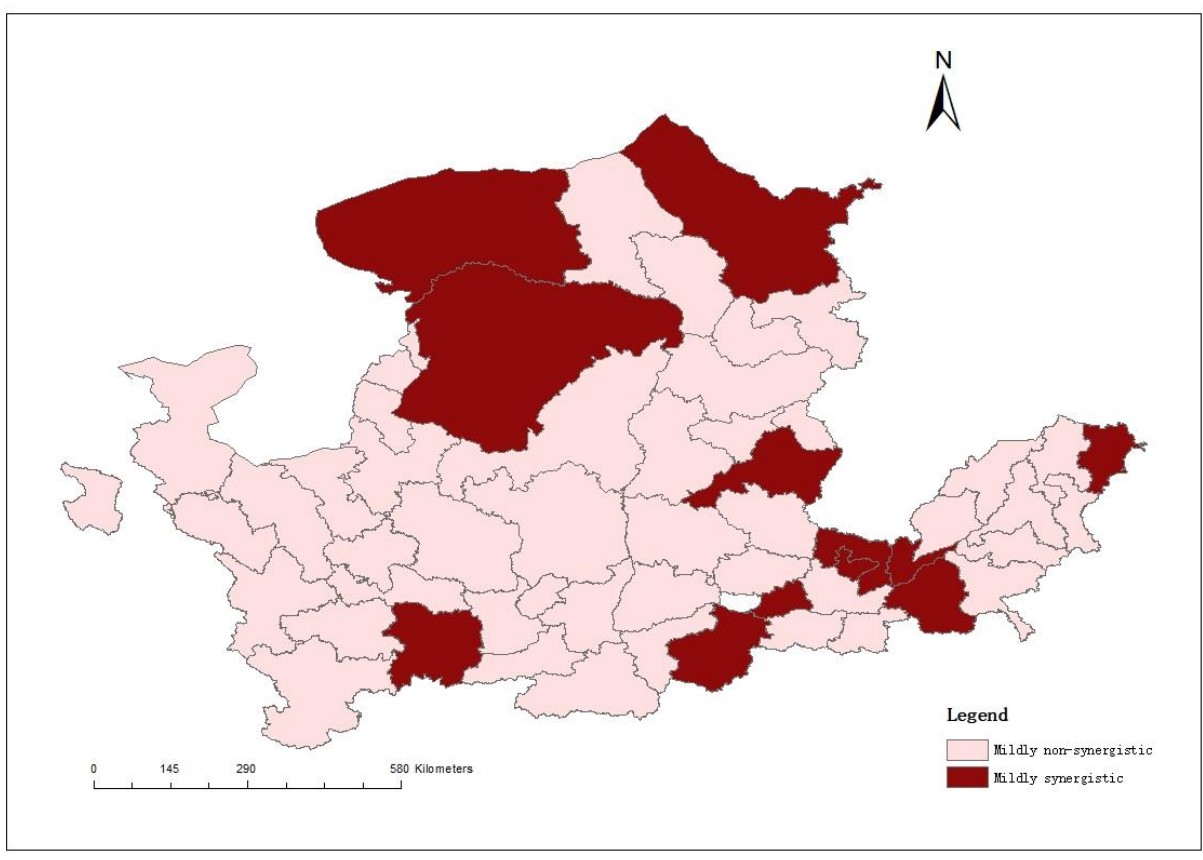

**Figure 5.** Spatial distribution of synergy degree in the Yellow River Basin in 2011.

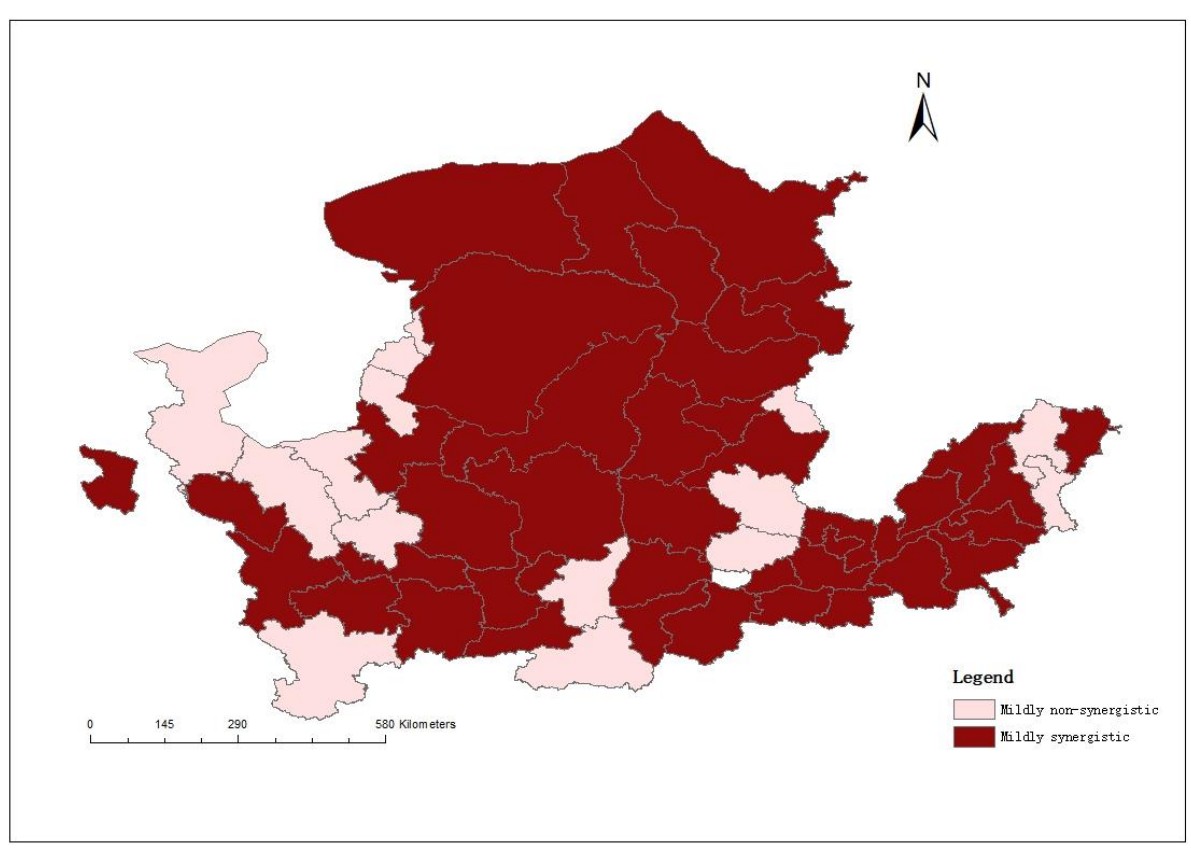

**Figure 6.** Spatial distribution of synergy degree in the Yellow River Basin in 2016.

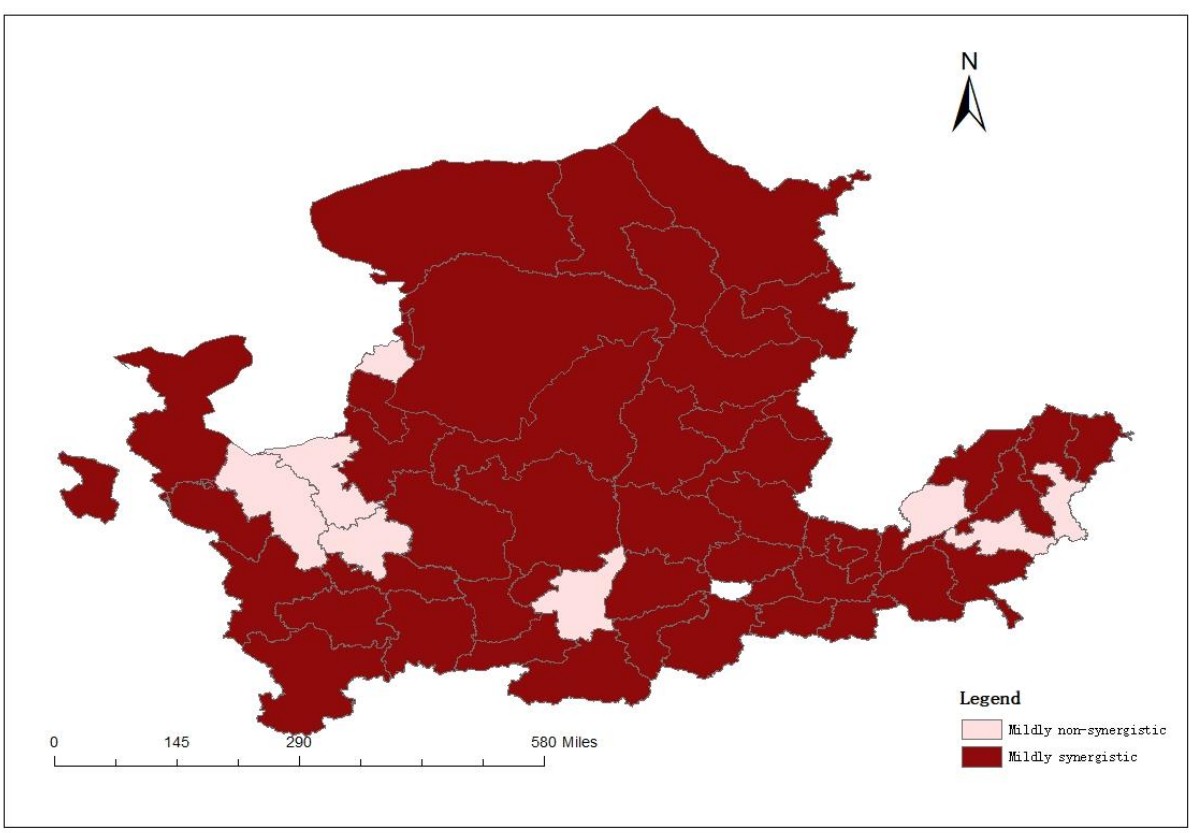

**Figure 7.** Spatial distribution of synergy degree in the Yellow River Basin in 2020.

4.2.2. Spatial Correlation Analysis

The analysis of the spatial distribution characteristics of synergy degree in the Yellow River Basin can only understand the level of synergy in each city, but cannot reflect its correlation, so it is necessary to reveal the global spatial autocorrelation by Moran's I index. In this paper, the global spatial autocorrelation of synergy is determined by constructing the neighboring spatial weight matrix and calculating the global Moran's I index using Geoda 1.20 software. The details are shown in Table 4.

**Table 4.** Moran's I Index of global environmental and industrial synergy degree in the Yellow River Basin from 2011 to 2020.

| Time | Moran's I | *p*-Value | Z Test Value | Standard Deviation |
|------|-----------|-----------|--------------|--------------------|
| 2011 | 0.0528 | 0.208 | 0.7842 | 0.0866 |
| 2012 | 0.1041 | 0.094 | 1.3745 | 0.0908 |
| 2013 | 0.1191 | 0.068 | 1.5764 | 0.0871 |
| 2014 | 0.0778 | 0.146 | 1.0575 | 0.0892 |
| 2015 | 0.0224 | 0.314 | 0.4228 | 0.0896 |
| 2016 | 0.1957 | 0.012 | 2.3985 | 0.0884 |
| 2017 | 0.1402 | 0.053 | 1.6600 | 0.0928 |
| 2018 | 0.0871 | 0.141 | 1.0906 | 0.0925 |
| 2019 | 0.1504 | 0.037 | 1.8840 | 0.0882 |
| 2020 | 0.1335 | 0.065 | 1.6374 | 0.0898 |

The global Moran's I index of the synergy degree between environment and industry in the Yellow River basin from 2011 to 2020 is greater than 0. However, the global Moran's I index of 2011, 2014, 2015 and 2018 do not pass the significance test, indicating that the synergy degree in these years has a scattered distribution. The remaining years all pass the 10% significance level test, and the synergy degree has a more significant spatial correlation characteristic. Looking at the index values that passed the test, it can be seen that the spatial

clustering of synergy degree in the Yellow River basin increases with time. In general, there is a strong spatial clustering of synergy degree in the basin, and the regions are more closely connected with each other, and the level of synergy can be improved by playing the leading role of nodes or central cities.

## 5. Conclusions

Based on the compound system synergy model, this paper measures and analyzes the synergy degree between environment and industry in the Yellow River Basin from 2011 to 2020. The calculation results show that, firstly, the synergy degree between environment and industry in the Yellow River Basin has been improved, but the level is still low and there is much room for development; there are significant temporal and regional differences in the synergy degree in the upper, middle and lower reaches of the basin and among cities. Second, the number of cities in the Yellow River Basin with mild synergy is increasing, which confirms that the overall environment and industry in the basin are evolving in a more synergistic direction; there is a global spatial positive correlation between the environment and industry synergy degree in the Yellow River Basin.

From an overall perspective, the synergy between environment and industry in the Yellow River Basin is at a low level, indicating that the long-term efforts of government, enterprises and residents are needed to achieve a good synergy between environment and industry in the basin; from a basin perspective, the synergy differs among basins, and it is necessary to realize the synergy between environment and industry in each basin according to local conditions, and the difference in synergy between basins requires the establishment of a synergistic development mechanism within the basin to achieve basin-wide synergy. From the spatial point of view, the global spatial positive correlation of synergy indicates that cities are closely connected and can play a leading role as important cities. Therefore, in order to realize the synergy between environmental protection and industrial development in the basin, and ultimately achieve high-quality development and sustainable development in the Yellow River Basin, this paper proposes the following recommendations.

(1) The government should form institutional constraints on secondary industry enterprises and promote green development of secondary industry. The important source of environmental pollution in the Yellow River Basin is the secondary industry, which needs to be focused on. One of them is to establish a negative list. Clearly set out the types of projects prohibited from investment and construction; clearly set out the list of prohibited and eliminated industries; strictly control wastewater, waste gas, solid waste emissions and serious overcapacity industries; and strictly control the new coal power and coal chemical industry, to force enterprises to green transformation. Second, strengthen environmental constraints. Through strict implementation of energy-saving and environmental protection laws and regulations, improve the level of environmental regulation and guide the Yellow River Basin around the city enterprises pay attention to green production. Enterprises should develop green industries. They should continue to introduce and develop green technologies and apply and promote them to achieve green production methods, such as introducing technologies to enhance waste treatment rates, and strive to transform the economic growth mode of the basin from resource-intensive development with high inputs to intensive development and transformation from pollution-led enterprises to clean industry-led; they should also develop new industries and move toward developing sustainable industries, such as new energy industries, high-end equipment manufacturing, and new energy industry, high-end equipment manufacturing and new generation information technology and software and other strategic new industries. Residents should improve their environmental protection awareness and knowledge base, and implement environmental protection awareness in reality.

(2) Combining the comparative advantages of the region, the strategic industries for the development of synergistic development of environment and industry in the upper, middle and lower reaches of the basin should be developed according to local conditions. The upstream source area should focus on water conservation and protection, but it also

needs to develop industries, and at the same time should create ecological products, so that environmental protection and industrial development can be synergized; the midstream environment and industry synergy level has reached a high level in the basin, but the reality of its large environmental pollution also requires it to maintain development and protection at the same time, and to develop industries that are ecologically appropriate to the basin. The downstream cities have a high economic level and need to continue to shift to new dynamics of development and accelerate the digestion of changes brought about by the new dynamics shift, as well as to strengthen the prediction and monitoring of floods and prevent flooding.

(3) Establish a synergistic development mechanism with interactions, interconnections and clear rights and responsibilities in the basin, and realize coordinated docking between the upper, middle and lower reaches of the basin. Taking into account the resources of the whole basin, specific plans as well as corresponding promotion measures are formulated to finally guarantee the synergy of the basin to achieve high-quality development, inter-regional flow of factors, and reduction in regional differences, so as to realize the synergistic and linked development of the whole basin. For example, we encourage water rights trading, i.e., under the premise of defining the initial water rights, we encourage water rights trading among different industries and different cities in the Yellow River basin, cultivate water rights trading market, and establish inter-regional cooperation and compensation mechanism.

(4) Strengthen the construction of urban agglomerations in the Yellow River Basin and lead the central cities to play a radiating role. The results of environmental and industrial synergy degree measurement of cities in the Yellow River Basin show that the central cities tend to have a high level of synergy degree in the basin, so we should accelerate the construction of urban agglomerations with the central cities in the basin as the core, and continue to develop the Guanzhong Plain Urban Agglomeration with Xi'an as the center, the Central Plains Urban Agglomeration with Zhengzhou as the center, and the Shandong Peninsula Urban Agglomeration with Qingdao and Jinan as the center, so that the strong major cities can The city clusters will play a leading role, so that their technology and talents can play a spillover effect and cultivate important economic growth poles.

**Author Contributions:** W.X.: Conceptualization, Methodology. Y.L.: Investigation, Writing—original draft and editing. All authors have read and agreed to the published version of the manuscript.

**Funding:** This work was supported by the Key Project of Natural Science Foundation of Shaanxi Province, China (No.2022JZ-41) and the National Natural Science Foundation of China (No.72273103).

**Institutional Review Board Statement:** Not applicable.

**Informed Consent Statement:** Not applicable.

**Data Availability Statement:** The data used to support the findings of this study are available from the corresponding author upon request.

**Conflicts of Interest:** The authors declare no conflict of interest.

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
