# Peer review of "Measurement of Synergy Degree between Environmental Protection and Industrial Development in the Yellow River Basin and Analysis of Its Temporal and Spatial Characteristics"

_sustainability, doi:10.3390/su15043386_

Round 1

Reviewer 1 Report

This article is devoted to very relevant issues related finally to the improvement of the environment, which is a guarantee of a high level of quality of life. The article is definitely scientific in nature, interesting, full of methods, sufficiently original and worthy of publication. At the same time, reading the text of the article gives grounds

Neither the introduction nor the abstract clearly states what the purpose of the conducted research is, for whom its results are intended, and what their practical significance is. One can only guess that the authors wanted to find an answer in their research to the question "whether the environmental and industrial compound system in the Yellow River Basin meets the synergetic conditions on the basis of constructing the compound system." But this cannot be the purpose of the research, because the answer to this question can only be the basis for further conclusions and recommendations. The recommendations made in the conclusions are somewhat too general and, in the end, in the vast majority, they can be made without conducting such complex calculations as made in this study.

The expediency of the formal allocation and assignment of structural unit 2 «Synergistic mechanism between environment and industry in the Yellow River Basin»  needs some explanation. Is it related to  methodology or results?

There are certain comments on the content of the illustration in Fig. 1. In the presented variant of the scheme, the interaction between the environment and industry is represented rather "hazy" - Nonlinear interaction. In reality, the interaction between these spheres is based on the triune role of the environment as a source of resources for industry, as a waste disposal site, and as a recreation site. The author mentions these things very generally as "complex positive and negative feedbacks between environmental and industrial subsystems".

The authors of the article usually have a better idea of what they wanted to research and what to get. But in my opinion, in this article it is more possible to talk about the sustainable development of the region than about synergy, since synergy involves obtaining a synergistic effect, and this is practically not mentioned in the article, with the exception of "1+1>2", which did not obtain further development in the article. At the very least, the results obtained in this study can be used for future studies of sustainable development in this region. 

Author Response

Dear reviewer 1,

    Thank you for your comments,please see the attachment for the detail of the modifications.

Reviewer 2 Report

GENERAL COMMENTS 

The manuscript entitled “Measurement of synergy degree between environmental protection and industrial development in the Yellow River Basin and analysis of its temporal and spatial characteristics" investigates the application of the dissipative structure theory and synergy concepts and modelling, to explain mechanisms between industry and ecology at the Yellow River Basin in China. 

The manuscript focuses on the concept of order and how coordination and cooperation work as linking elements and lack of them are expresses in terms of entropy, in the system industry-environment analysed, however, some basic components are absent, and this is important because the type of industry determines the kind of emissions, as the atmospheric emissions declared only as being of sulphur dioxide, is the emitter a coal power plant (using pyritic coal, for example)? Or the type of solid waste discharged, are these mostly organic, plastic, inorganic, toxic? A bit more of detail on the industry could justify what is loosely listed in Table 2.

 It will be desirable that authors explain how the system industry-environment is isolated of the impact of other components, from society or economy. Undoubtedly, the covid19 lockdown regulations have had an impact of the system under study, at least for the year 2020.

Author Response

Dear reviewer 2,

    Thank you for your comments,please see the attachment for the detail of the modifications.
